# Evaluating Ecologically Acceptable Sprout Suppressants for Enhancing Dormancy and Potato Storability: A Review

**DOI:** 10.3390/plants10112307

**Published:** 2021-10-27

**Authors:** Nyasha Gumbo, Lembe Samukelo Magwaza, Nomali Ziphorah Ngobese

**Affiliations:** 1Department of Botany and Plant Biotechnology, University of Johannesburg, P.O. Box 524, Johannesburg 2006, South Africa; nyashagumbo@rocketmail.com; 2Discipline of Horticultural Science, School of Agricultural, Earth and Environmental Sciences, University of KwaZulu-Natal, Private Bag X01, Scottsville 3209, South Africa; Magwazal@ukzn.ac.za

**Keywords:** postharvest storage, shelf-life, sprout inhibition, potato

## Abstract

Postharvest losses are a key stumbling block to long-term postharvest storage of potato tubers. Due to the high costs and lack of infrastructure associated with cold storage, this storage method is often not the most viable option. Hence, sprout suppressants are an appealing option. In most developing countries, potato tubers in postharvest storage are accompanied by a rapid decline in the potato tuber quality due to the physiological process of sprouting. It results in weight changes, increased respiration, and decreased nutritional quality. Therefore, proper management of sprouting is critical in potato storage. To avoid tuber sprouting, increased storage and transportation of potatoes demands either the retention of their dormant state or the application of sprout growth suppressants. This review evaluates the current understanding of the efficacy of different sprout suppressants on potato storability and the extension of potato shelf-life. We also consider the implications of varied study parameters, i.e., cultivar, temperature, and method of application, on the outcomes of sprout suppressant efficacies and how these limit the integration of efficient sprout suppression protocols.

## 1. Introduction

Potato (*Solanum tuberosum* L.) is consumed by many people worldwide, with a global monthly consumption of potato per capita of 31.3 kg as of 2018 [1]. The total global potato production exceeds 300 million metric tons every year. The Food and Agriculture Organization (FAO) has strongly endorsed potato as a food security crop since the world is confronted with inadequate food supplies, increased population growth, and food demand [1]. Potato is a staple food, regarded as an essential commodity in global nutritional security [2,3]. Given its large yield and excellent nutritional content, it is an essential food-security crop and cereal crop alternative [2]. As a result of the above, improving and ensuring the quality and storability of potatoes after harvest is crucial for economic and food security reasons. 

Sprouting is one of the most significant challenges in the postharvest storage of potato tubers and throughout the entire supply chain, as it reduces the quality and quantity of marketable produce, thereby resulting in financial losses [4,5,6,7]. Severe losses are incurred due to potato tuber sprouting and sprout growth since these cause alterations in tuber physical properties, such as reduced turgidity, induced shrinkage, and fosters weight loss [8,9]. Sprouting also leads to the accumulation of toxic compounds in the potato flesh, such as solanine and chaconine [9], and a reduction in nutritional and processing qualities [10]. Potato tubers are mainly consumed fresh, resulting in a yearly demand, and necessitating extended postharvest storage of tubers after harvest. Currently, common strategies for long-term storage of potato tubers include storage at low temperatures between 2–4 °C (90–95% relative humidity) or between 8–12 °C (at 85–90% relative humidity) and/or the use of chemical compounds that act as sprout suppressants [11]. Long-term storage at low temperatures, on the other hand, degrades potato quality [12,13]. A wide range of sprout inhibitors can be employed to prevent these incidents, which may result in significant economic losses to tuber producers if ignored [14,15].

Globally, isopropyl *N*-(3-chlorophenyl) carbamate (CIPC; chlorpropham) is the most used potato sprout suppressant among commercial potato producers. Although CIPC is a very efficient sprout suppressant, its continuous use is actively being discouraged because of safety concerns. It has been shown to be detrimental to both the environment and consumer health [11,16]. For instance, the degradation products of CIPC, such as aniline-based derivatives, e.g., 3-chloroaniline, have been reported to be pollutants that are highly carcinogenic and toxic to the environment [17,18]. These toxicological and other concerns have led different countries, notably the European Union, to progressively regulate and, in some cases, completely prohibit the use of CIPC [11]. The toxicological and environmental risks of CIPC use necessitate the development and adoption of novel sprout suppressing compounds that are safer for humans and environmentally friendly. Several research efforts have been targeted at exploring and exploiting the sprout suppressing qualities of various chemical compounds to discover ecologically acceptable alternatives [15].

Finding a suitable sprout suppressant that can match the efficiency of CIPC has been quite daunting. Several promising alternatives have been identified. For instance, *S*-carvone is a naturally occurring monoterpene that inhibits potato sprouting [9,19,20,21]. Other promising compounds with significant potato sprout suppression properties include 1,4-dimethyl naphthalene [9,22,23], maleic hydrazine [24], and 3-decen-2-one [9]. Essential oils, and chemical components of essential oils such as monoterpenes, have also been tested and used as suppressants of sprouting in potato tubers with a significant level of efficacy [7,10,25].

Critical parameters that must be taken into consideration in the evaluation of chemical compounds for use as potato tuber sprout suppressants for the extension of dormancy and tuber storage management include the type of cultivar, chemical nature and bioactivities of the compound, dosage, storage temperature, and mode of application, among others [25,26,27]. Several studies have evaluated these parameters to determine how they influence the efficacies of sprout suppressants [4,9,24,26,28]. However, variations in these parameters among different studies reported in literature impede comparisons of the efficacies of different candidate sprout suppressants and the integration of these research findings to inform decisions on which of these novel sprout suppressants outperform the other.

This review provides a critical but concise overview of different candidate chemical compounds that have shown potential bioactivities and significant efficacies for use as potato tuber sprout suppressants and potential alternatives to CIPC. We highlight the variations in values of critical parameters, such as temperature, treatment dosage, and cultivar types, observed in reported studies of alternative suppressants in the extension of shelf-life of potato tubers. This is to provide answers to the following essential questions: (1) How do experimental data from shelf-life studies and the efficacies of different sprout suppressants measure up in helping to inform users’ choice of the best performing suppressants? (2) Do data provide sufficient experimental evidence on the implications and impacts of tuber storage conditions, especially temperature, on the efficacies of sprout suppressants to enable comparison of suppressant efficacies for best tuber sprouting management protocols? (3) What can be gleaned from existing experimental data on the implications of suppressant application methods, dosage, and potato cultivars on the efficacies of the evaluated sprout suppressants? (4) What research gaps exist and what research directions should be charted to discover and develop natural, non-toxic, and eco-friendly alternatives to CIPC? The implications of these on the integration of data from these studies to compare suppressant efficacies, the development of alternatives to CIPC, and informing the application of these for the extension of tuber shelf-life and storage management are also discussed.

## 2. Naturally Occurring and Ecologically Safe Tuber Sprout Suppressants

### 2.1. 1,4-Dimethyl Naphthalene

1,4-dimethyl naphthalene (1,4-DMN), a naturally occurring and endogenous methyl-substituted naphthalene in potatoes, is an alternate sprout inhibitor [22,23,29]. It is a volatile compound that contributes to the flavor and aroma of baked potatoes [23] and was isolated from potato skins and then synthesized for use as a plant growth regulator [30]. In particular, the chemical suppresses sprout production and etiolated development in stored potato tubers, thereby prolonging the effective storage period and preserving tuber quality [22,29]. Because the chemical has reversible effects, it may also be utilized on seed potatoes [23]. 1,4-DMN is commercialized in synthetic form as 1,4Sight^®^, 1,4SHIP^®^, and 1,4SEED^®^.

#### 2.1.1. Mode of Action of 1,4-Dimethyl Naphthalene

Meigh, et al. [31] demonstrated the availability of 1,4-DMN isomers and how they exhibit sprout-inhibiting properties. Studies conducted by [32] revealed the potential of 1,4- and 1,6-DMN to reduce the rate at which potato sprouting occurs, and these findings were confirmed by [33]. The mechanism of action of 1,4-DMN is yet to be fully characterized. However, because it is a naturally occurring substance that is readily available in potato tubers, it is thought to suppress sprout development by extending endogenous dormancy conditions and via hormonal actions [22,23]. Although emerging, reports suggest that 1,4-DMN inhibits sprouting by repressing meristem cell proliferation [22,29]. Analysis of the changes in transcriptional profiles of meristems isolated from 1,4-DMN- treated potato tubers showed the repression of cyclin or cyclin-like transcripts, thus suggesting that 1,4 DMN modifies genes involved in the maintenance of a G1/S phase block, most likely via the stimulation of the cell cycle inhibitors [22]. A recent report shows that sensitivity to 1,4-DMN changes as potato tubers age and transition from endo-dormant to eco-dormant in storage [29]. These are clear indications that 1,4-DMN may regulate sprouting by integrating external/ambient cues.

#### 2.1.2. Evaluation of 1,4-Dimethyl Naphthalene as a Sprout Inhibitor

The efficacy of 1,4-DMN as a sprout inhibitor has been the subject of much of the published studies available in the public domain. Many of these studies have indicated the efficacy of the 1,4-DMN based on how long the experiments ran for, which is the storage period, rather than how long 1,4-DMN was able to extend and suppress sprouting (which would be shelf-life extension). For instance, a study by Kalt, Prange, Daniels-Lake, Walsh, Dean and Coffin [4] revealed that a dosage application of 0.02 mL/kg of 1,4-DMN did not result in any significant shelf-life extension. These are shown in Table 1. Compared to CIPC, Russet Burbank cultivars did not achieve any shelf-life extension. In addition, controls were not used in this study. Therefore, it becomes difficult to allow for better comparison with other studies conducted using controls or both. A recent study by Nyankanga, Murigi, Shibairo, Olanya and Larkin [28] used both Control and CIPC. They demonstrated that a dosage application of 0.1 mL/kg of 1,4-DMN could achieve a shelf-life extension of 10 days and 18 days compared to the control. These were achieved using Asante and Kenya Mpya cultivars, respectively. Treatment with 1,4-DMN did not result in any significant shelf-life extension when the Shangi cultivar was used. However, when compared to CIPC, none of the three cultivars could achieve shelf-life extension at all.

As shown in Table 2, several studies have demonstrated the efficacy of 1,4-DMN based on how long different varieties of stored potato tubers maintain their quality. Baker [34] showed that the Russet Burbank variety was stored for up to 330 days with a 0.2 mL/kg dosage of 1,4Sight^®^ at 7–8 °C. This dosage tends to be more efficient when compared to the results from other studies. In contrast, de Weerd, Thornton, and Shafii [26] demonstrated stored potatoes could be maintained for just 66 days at a higher temperature of 15 °C +/2 °C with a lower dose of 0.056 mL/kg. This shows that a higher dosage concentration worked better at increasing the storage period of potato varieties. Discrepancies in some important parameters exist in these two studies. While Baker [34] employed a swing fogger to apply a higher dosage, de Weerd, Thornton, and Shafii [26], with the least performing dosage, used a gauze to apply the same chemical. Overall, it is challenging to decide the best potential sprout suppressant alternatives since there are so many variations in critical parameters that were considered, such as dosage application, temperature, method of application, and what genetic type of potatoes is used. These could have contributed to the significantly different outcomes of both studies.

Potato varieties were stored for up to 200 days at 4 °C at a dosage of 0.04 mL/kg as conducted by Richard Knowles, Knowles, and Haines [23]. However, the suppressant dosage used was lower than the dose (0.1 mL/kg) used in the study conducted by Beveridge, Dalziel, and Duncan [35], where potato tubers were only stored for 98 days at a higher temperature of 10+/0.5 °C.

Lewis, Kleinkopf, and Shetty [30] evaluated 1,4-DMN, diisopropyl naphthalene (DIPN), and CIPC for reducing sprouting in Russet Burbank potatoes and discovered that DIPN was the most efficient of the two naphthalene derivatives when two applications of the suppressant were used. They found that 1,4-DMN or DIPN was an effective sprout suppressant on a short-term basis, which is very high compared to the rates recommended. The study conducted by Baker [34] demonstrated that the application of 1,4-DMN at 0.06 mL/kg was as effective as CIPC. Sensory detection threshold levels for residual levels of 1,4-DMN in Russet Burbank potatoes treated with 1,4-DMN were low, and 1,4-DMN did not induce significant changes in sensory quality of stored potatoes compared to CIPC [36].

### 2.2. 1,4 SIGHT^®^

1,4-DMN has acquired registrations for use in different European countries as of 2018. In that way, the synthetic form has been marketed with the trademark 1,4Sight^®^ [37], among others. On short-dormancy potato varieties, 1,4SIGHT^®^ can be applied as a stand-alone to maintain dormancy (inhibit sprouting) and quality while keeping moisture loss at a bearable minimum immediately postharvest.

#### 2.2.1. Mode of Action of 1,4SIGHT^®^

1,4SIGHT^®^ is a ‘therapy’ that is based on genetics. It regulates genes involved in water-holding proteins, which may aid in weight reduction [38]. Pathogen resistance genes are also regulated, resulting in greater resistance to fungal infection [38,39]. The method of action of 1,4SIGHT^®^ is fungistatic, which means that the fungus is prevented from growing, allowing non-pathogenic bacteria and fungi to proliferate [40].

#### 2.2.2. Evaluation of 1,4SIGHT^®^ as a Sprout Inhibitor

1,4Sight^®^ has been used in a shelf-life study conducted by Kalt, Prange, Daniels-Lake, Walsh, Dean, and Coffin [4], where extension of shelf-life was not achieved at all compared to CIPC (Table 1). However, there is a greater chance that shelf-life would have been achievable with a control other than CIPC. In Table 2, another study conducted by Baker [34] showed the most extended storage period of 330 days at a dosage application of 0.2 mL/kg at 7–8 using a Swing fogger apparatus. However, this was not a shelf-life extension study. Studies that compare the effects of varying temperatures and modes of application on the efficacy of 1,4Sight^®^ are not available.

### 2.3. S-carvone

Plant organs, such as leaves, roots, stems, and flowers, contain high concentrations of essential oils. Volatile oils, also known as ethereal oils, obtain their names from their ability to evaporate quickly when exposed to air at room temperature. Secondary metabolites, such as sesquiterpenes and phenylpropanoids, make up most of these oils. They are well-known for their antimicrobial and sprout-inhibiting properties [41]. Both *S*-carvone, 2-methyl-5-(1-methylethenyl)-2-cyclohexene-1-one, and its enantiomer, *R*-carvone, are volatile monoterpenes in the essential oils of caraway (*Carum carvi* L.), mint (*Mentha spicata* L.), and dill (*Anethum graveolens* L.), which have potent inhibitory bioactivities on the sprouting of potato tubers at continuous low headspace concentrations [8,42,43]. In addition to its sprout suppression bioactivities, *S*-carvone inhibits bacterial and fungal growth, thereby presenting secondary benefits, such as suppressing storage pathogens such as *Fusarium* and *Rhizoctonia* species [43,44]. Other notable advantages of *S*-carvone over CIPC include its strong odor, which is transmitted to foods when used as a flavoring agent, it is non-toxic and safe for humans, and it contributes less to ozone depletion compared to CIPC [44,45]. Some European nations have commercialized *S*-carvone and market it under tradenames such as Talent^TM^ [27].

#### 2.3.1. Mode of Action of *S*-carvone

The precise mechanism of sprout suppression employed by *S*-carvone is yet to be fully resolved. However, *S*-carvone is believed to influence potato tuber sprouting by interfering with isoprenoid metabolism. The mevalonate pathway, which employs the enzyme 3-hydroxy-3-methylglutaryl coenzyme A reductase (HMGR), is implicated in the process that prevents sprouts from growing [20]. *S*-carvone interferes with sprouting by inhibiting HMGR activity [46] through repression at the post-translational level [47]. Another model proposes the inhibition of the 2-C-methyl-D-erythritol 4-phosphate (MEP) isoprenoid pathway, which affects the mevalonate pathway downstream and isoprenoid metabolism by blocking protein isoprenylation. Here, *S*-carvone blocks an MEP pathway-dependent protein geranylgeranylation that is required for signaling [48]. The mevalonate pathway partakes mainly is the provision of metabolites for the biosynthesis of hormones that are important for plant growth.

#### 2.3.2. Evaluation of *S*-carvone as a Sprout Inhibitor

Hartmans, et al. [49] applied *S*-carvone at a dosage of 0.6 mL/kg to two different cultivars, as shown in Table 3. Sprout growth inhibition was achievable for the Bintje cultivar only for 15 days compared to CIPC and the Agria cultivar, 0 days compared to CIPC. Sprout suppression for treated Russet Burbank cultivar was achievable for 70 days with a dosage of 0.080 mL/kg [4]. An in vitro study demonstrated the effectiveness of the essential oils and clearly showed that sprout growth and extension of shelf-life were achievable [50]. Using mint (*M. spicata*) essential oil, which contains a significant amount of carvones (51–73%) [8,21], and synthetic *R*-carvone, Teper-Bamnolker et al. [8] noted a significant decrease in sprouting and weight loss in tubers of eight different potato cultivars that were stored for six months. However, these studies were not shelf-life extension studies as they only demonstrated the effectiveness of the essential oil at reducing sprout growth.

With a dosage application of 0.6 mL/kg, sprout suppression for the Monalisa cultivar compared to the control was achieved 21 days [51]. Dosage application of 155 mL/kg extended tuber shelf-life by 25 days compared to the control for both Agria and Kennebec cultivars [20]. Using the Agria cultivar, [27] demonstrated that the tuber shelf-life extension at different temperatures was achieved with a dosage application of 0.6 mL/kg. Compared to the control and CIPC, different results were achieved with *S*-carvone. CIPC performed better than *S*-carvone at shelf-life extension. At 5 °C, they noted 60 days of the shelf-life extension was achieved compared to the control, whereas 0 days compared to CIPC. At 10 °C, 75 days extension was noted compared to the control, whereas 0 days was achieved compared to CIPC and at 15 °C, 90 days compared to the control and 15 days compared to CIPC.

### 2.4. SmartBlock^®^

SmartBlock is a biopesticide, i.e., it is a naturally occurring chemical with minimal detrimental environmental impacts. The active compound in SmartBlock^®^ is 3-decen-2-one, a naturally occurring 10-carbon unsaturated ketone [52] that has been tested on several potato species and under a variety of storage settings. Along with other *α,β*-unsaturated ketones, 3-decen-2-one is produced in higher plants as components of their aroma profiles [53]. Many industrialized nations have accepted and approved 3-decen-2-one as a food additive and flavoring agent in different processed foods. SmartBlock^®^ is intended for use in thermal fogging systems in potato storage facilities, especially for fresh market potatoes. It delivers a safe, quick sprout burn-off on fresh potato types without harming potato quality, and it is simply administered using fogging equipment [11].

#### 2.4.1. Mode of Action of SmartBlock^®^

For regulating postharvest sprouting in potatoes, SmartBlock^®^ has a unique mode of operation known as sprout ‘burn out’. When used as a hot or cold fog, the active 3-decen-2-one vaporizes quickly and easily, destroying the meristematic tissues of rapidly developing sprouts [9,52]. These *α,β*-unsaturated ketones are electrophiles, with carbonyl and conjugated double bonds, forming adducts with cellular amino and sulfhydryl groups, such as those in glutathione, proteins, and DNA, which is toxic and lethal to tissues [52]. 3-decen-2-one is also known to induce the disruption of internal cell structures and cell content leakage, interference with oxidative stress control, and rapid desiccation of sprouts [9,52,54].

Another notable mechanism of action of SmartBlock^®^ is the induction of a transient increase in respiration that mobilizes available reducing sugars before tuber respiration rate is decreased to similar levels as observed in dormant, non-sprouted tubers [54]. Sprout control bioactivities are also present in the first two breakdown products, 2-decanone and 2-decanol, which together provide extended sprout control. According to data, fresh market potatoes stored at colder temperatures (3–4 °C) can be safely stored with only one application during the storage season. For processing potatoes, which are usually stored at higher temperatures (7–10 °C), two–three applications are typically needed during the storage season [37].

#### 2.4.2. Evaluating SmartBlock^®^ as a Sprouting Inhibitor

According to the European Food Safety Authority [55], SmartBlock^®^ has shown the potential to be used as a sprout inhibitor (Table 4). Immaraju and Zatylny [56] demonstrated that using SmartBlock^®^ for successful sprout suppression for 21 days, at an application rate of 0.115 g/kg, with only one treatment, and at a higher temperature, is feasible for already sprouted potatoes. They noted that 100% of sprout eyes were burnt off until the last day of observation, whereas 94% of potato studies were blackened. SmartBlock^®^ can be perceived as a very effective sprout inhibitor as it can perform well at even higher ambient temperatures. The active component of SmartBlock^®^ (3-decen-2-one) has been suggested as a valuable alternative to CIPC for controlling sprouting in potato tubers [9].

The outcome of a recent study on the efficacy of SmartBlock^®^ in sprout inhibition indicated that this chemical is viable for use on various cultivars [54]. It was noted that multiple applications (three times) were required for effective sprout inhibition at higher temperatures whereas, for a lower temperature, only one application was enough to suppress sprouting for 168 days. When fresh potatoes are stored at 4 °C, a single application of dosage, ranging from 0.100 mL/kg to 0.135 mL/kg can provide season-long sprout control for many varieties. Processing varieties stored at 7.5 °C would require three applications.

### 2.5. Caraway Seeds and Essential Oils as Alternative Sprout Suppressants

Caraway seeds were used to inhibit sprout formation in the Monalisa cultivar (Table 5). Using seed essential oils, sprout suppression was achievable for 25 days at a 155 mL/kg dosage application in both Agria and Kennebec cultivars [20]. Similarly, at 5 °C storage temperature, dill essential oil suppressed sprouting by 90 days compared to the control whereas 30 days compared to CIPC [27]. At 10 °C, 135 days of the shelf-life extension was achieved with dill essential oil compared to the control, while 60 days was achieved compared to ClPC. Lastly, at 15 °C, 150 days of the shelf-life extension was achieved with dill essential oil compared to the control and 75 days compared to CIPC.

The potato tuber sprouting suppression bioactivities of essential oils is partly attributed to the abundance of diverse monoterpenes in these oils [8,20,25,57]. Monoterpenes are known to compromise membrane integrity because of their lipophilic nature [58]. Mint essential oil induced tuber bud necrosis by damaging apical meristem and vascular tissues [8]. Monoterpenes may influence phytohormones synthesis and activities to elicit sprouting suppression. For instance, 1,8-cineole-mediated inhibition of tuber sprout growth was found to be mediated via the alteration of key gibberellin metabolism gene expression, impaired gibberellin biosynthesis, and reduced gibberellin content [10]. Other essential oils with reported potato sprout suppression activities include those obtained from eucalyptus and coriander [20,57].

With essential oils, several treatments are necessary during storage to sustain sprouting inhibition, and because the essential oil manufacturing process is quite expensive, these types of sprout suppressants are challenging to put on the market [59]. However, compared to CIPC, essential oils provide no difficulty when storing potato seeds in the same facility as the treated potatoes since their impact is reversible, and their volatility makes it easy to clean the storage facility’s air of any chemical residues [25]. They also provide secondary benefits as they can diminish the rate of accumulation of reducing sugars in stored tubers, which are responsible for browning in processed potato products [7]. Another important consideration for promoting and adopting essential oils, or their components, is their safety. Since these compounds are from natural sources and biodegradable, they are safe for human consumption and do not pose any threat to the environment. Using essential oils will also encourage the cultivation of plants from which they are extracted, thereby contributing to job provisions and the agricultural economy.

### 2.6. Aloe Vera Gel

Due to its unique nutritional profile, Aloe vera is extensively used in the food, health, and nutraceutical sectors. As an edible coating, Aloe vera gel has grabbed the curiosity of researchers who wish to look at its potential for increasing the shelf and storage life of fresh fruit due to its organic origin [60,61]. Edible coating is a preservative technology that involves the application of a thin layer of edible material, which may be hydrophobic or hydrophilic or an integration of both, around the farm produce to restrict respiratory gas exchange [62,63,64,65,66]. This increases carbon dioxide accumulation and decreases oxygen supply while limiting water loss, thus extending the storage life of fresh commodities [60]. Edible coating is gaining popularity for controlling the ripening of vegetables and climacteric fruits because it is easy to prepare, widely available, relatively inexpensive, and does not require the use of sophisticated instruments [60,61].

Extrapolating from results obtained in studies that used *Aloe vera* gel as edible coatings on fruits and vegetables, the potential outcomes and benefits of testing and adopting *Aloe vera* gel for use as a sprout suppressant for potato tuber storage can be glimpsed. Edible coatings made from *Aloe vera* gel have been found to prevent weight loss by reducing moisture loss and retaining fruit firmness, lowered respiration and delayed oxidative browning, and inhibit microbial growth in diverse fruits and vegetables [60,63,66]. However, there is hardly any data on the usage of *Aloe vera* gel as a sprout suppressant or its application on potato cultivars for tuber shelf-life extension.

## 3. Implications of Temperature, Cultivar Type, and Mode of Application on Sprout Suppressant Efficacies

### 3.1. Temperature

Temperature is an important environmental factor with far-reaching impacts on growth and developmental processes in plants. Several studies have demonstrated the impacts of low temperature and varying low temperature on tuber sprouting during storage [23,66]. These may be attributed to the fact that temperature is a potent modulator of many enzymes and proteins that mediate growth and developmental process. Generally, low temperatures, especially between 8–12 °C, are more favorable for the suppression of tuber sprouting during storage [54,66]. Combining low temperatures with sprout suppressant treatments has been shown to enhance the suppression of tuber sprouting and extended dormancy [66]. Although low-temperature storage after treatment with sprout suppressants is beneficial, it comes with extra cost and is difficult to achieve in tropical regions [28]. In addition, maintaining tubers at low temperatures may be even more difficult and expensive during long-distance transportation or export. These, therefore, necessitate the need to evaluate suppressants for their efficacies at high ambient temperatures. Generally, emerging data suggest that commonly used sprout suppressants, such as 1,4-DMN, 3-decen-2-one, and *S*-carvone, elicit lesser sprout suppression bioactivities at higher temperatures [27,28,66]. An approach to addressing this challenge would be to evaluate suppressant dosages, frequency of application, and the synergistic effects of different suppressants on tuber sprouting at high temperatures.

### 3.2. Cultivar Type

Genetic variability is a significant basis for the differences in crop cultivars. Moreover, the responses of different cultivars to chemical compounds are often shaped by their genetic make-up. With respect to sprouting, potato cultivars vary in the length of their dormancy periods. For example, recent observations with untreated tubers of Asante, Kenya Mpya, and Shangi that were stored at ambient temperature (23 °C) showed that while Asante and Kenya Mpya tubers sprouted after 8 weeks of storage, Shangi tubers only took 2 weeks to sprout [66]. In the same study, the same cultivars were treated with CIPC at a dosage of 100 mg kg^−1^ at 23 °C and observed, sprouting occurred on Kenya Mpya, Asante, and Shangi tubers after 16, 10, and 4 weeks, respectively. In contrast, tubers of all three cultivars did not sprout until after 20 weeks of storage when treated with 22 mg kg^−1^ of CIPC. Similar variations were observed in tuber sprouting responses in different cultivars treated with 1,4-DMN [28]. In an earlier study, cultivar-dependent responses to CICP treatment were noted under similar storage, suppressant dosage, and method of application [49]. Given these, the development and optimization of suppressant application regimes need to consider the potato cultivar to be treated. Unfortunately, very few studies factor this in their experimental designs.

### 3.3. Mode of Application

Adopted methods of application of agrochemicals, including sprout suppressants, exert a significant influence on the performance and efficacies of agrochemicals. As a result, the mode of application of sprout suppressant is a critical factor to be considered in the optimization of sprout suppression for postharvest potato storage. Common methods of application of sprout suppressants include cold or thermal fogging, aerosol spray, and vaporization of headspace in a sealed area [20,23,54]. The mode of application is partly determined by the physical and chemical nature of suppressants. For instance, volatiles, such as 1,4-DMN, 3-decen-2-one, *S*-carvone, and essential oils may be applied by fogging or vaporization in a tightly sealed space [20,54].

From existing reports, it is evident that the mode of application of sprout suppressants influences its efficacy. For instance, both Hartmans et al. [49] and Sanli and Karadogan [27] used the same potato cultivar, Agria, similar dosage (0.6 mL/kg), and worked at a similar ambient temperature of 5 °C to evaluate the efficacy of *S*-carvone in extending the shelf-life of stored potato tubers (Table 3). They obtained very different outcomes, which can be attributed to the disparity in the mode of application and the frequency of sprout suppressant treatments. While Hartmans et al. [49] used a swing fog apparatus, Sanli and Karadogan [27] used a wick freshener. Although other factors, such as frequency of suppressant application and the timing of application, may have also contributed to the different outcomes reported by these workers. Generally, there is a paucity of experimental data that focuses on how the mode of application of sprout suppressants affects their efficacy and performance.

## 4. Conclusions

Overall, because there are so many important endogenous and exogenous factors to consider when evaluating sprout suppressants’ efficacies, reaching a decision on the most effective sprout suppression alternative is quite difficult. Generally, a very high number of differences in study parameters was the major bane of a seamless comparison of the different studies to determine and rank the efficacies of each sprout suppressant. These differences in study conditions and parameters also resulted in the observation of differences even when different authors evaluated the same sprout suppressant. Therefore, this necessitates a study that evaluates the efficacies of different sprout suppressants with all the critical parameters, especially temperature, dosage, and application method, maintained constantly and with due considerations to cultivar differences. The quantity of material accessible on sprout suppressants that have been registered and commercialized based on shelf-life is minimal. It should be emphasized that the tables’ efficiency rankings were produced based on the findings and conclusions of the writers. These are estimates because the methodologies and critical parameters used in each experimental study vary. However, these were quite sufficient to allow for some level of comparison of the suppressant efficacies, although with great caution.

Furthermore, it is crucial to extend the search for naturally occurring and safe alternative suppressants that can replace CIPC to essential oils, their bioactive components, as well as Aloe vera gel. Although some essential oils and their components have been evaluated for their sprout suppressant properties, many of these studies do not evaluate the effects of differences in cultivar used. With regards to Aloe vera gel, we could find no report that evaluated it for use as a sprout suppressant despite its extensive use as an edible coating. Finally, we recommend that due research considerations should be given to the evaluation of critical suppressant use and application parameters, such as responses of individual cultivar to different sprout suppressants, analysis of suppressants at higher ambient temperatures, as well as how individual application regime and mode of application impact the efficacies of sprout suppressants.

## 5. Methodology

We adopted a descriptive and systematic review approach to address the objectives of this review. This review is descriptive since it seeks to determine and describe the extent to which the current body of knowledge on the efficacies of alternative sprout suppressants reveals any interpretable pattern or trend with respect to pre-existing propositions, theories, methodologies, or findings [67]. Moreover, since this review is targeted at synthesizing scientific data to reveal the current state of research on potential and natural sprout suppressant alternatives to CIPC, we adopted a systematic review approach. Systematic reviews aggregate, appraise, and synthesize extant empirical data and evidence that meet a set of previously specified eligibility criteria with a view to answering a clearly formulated and often specific research question [68,69].

### 5.1. Search Strategy

A semi-structured search strategy was carried out on indexed articles on scientific databases using specific keywords and phrases. The reference lists of included studies were hand-searched.

### 5.2. Study Selection, Inclusion, and Exclusion Criteria

For this review, we included studies that primarily evaluated and reported the efficacies of sprout suppressants, either the pure active compound or commercially available products, at extending the shelf-life and/or storage period of potato tubers. These also included studies that assessed the impacts of all or some of the following: cultivar types; storage conditions, especially temperature; suppressant concentration or dosage; as well as application protocols including frequency of suppressant application, stage of application, and method of application.

Due to the relatively limited number of experimental studies in potato sprout suppressant research focused on shelf-life and storage period extension, we did not subject our search to any time limits or timeline. We excluded studies that were conducted on sprouting suppressants but did not include assessments of shelf-life and storage period extension. The review was restricted to the English language.

### 5.3. Data Extraction

A data extraction table was designed and used to collect and record data on treatment, dosage, temperature, cultivar type, application protocol, shelf-life, and storage period from the selected studies.

### 5.4. Data Synthesis

High levels of data heterogeneity, especially with respect to variations in the combinations of parameters assessed among the selected studies, and an observable mixed data quality meant that statistical synthesis was impossible. We, therefore, adopted a combination of narrative and descriptive approaches to summarize our findings.

## Figures and Tables

**Table 1 plants-10-02307-t001:** Shelf-life studies showing the efficacy of 1,4-DMN and 1,4Sight^®^ as alternative sprout suppressants.

Treatment	Dosage	Temp.	Type of Cultivar	Application			Shelf-Life Extension (±) + Extended − Did Not Extend	Ref.
				Number	Stage	Method		
1,4Sight^®^	0.02 mL/kg	9 °C	Russet Burbank	1 Repeated after 9 weeks	After curing	Applied as an aqueous spray	Russet Burbank −70 days compared to CIPC	[4]
1,4-DMN	0.1 mL/kg	23 °C	Shangi Asante Kenya Mpya	1	After curing	Liquid fog	Asante +10 days compared to control. −70 days compared to CIPC. Kenya Mpya +18 days compared to control. −48 days compared to CIPC. Shangi 0 days compared to control. −105 days compared to control.	[28]

**Table 2 plants-10-02307-t002:** Efficacy studies on 1,4-DMN and 1,4Sight^®^ as alternative sprout suppressants.

Treatment	Dosage	Temp.	Type of Cultivar	Application			Storage Period	Ref.
				Number	Stages	Method		
1,4-DMN	0.1 mL/kg	10 ± 0.5 °C	Record Redskin Maris Peer Red Craigs Royal	1	After curing	Alumina carrier	98 days	[35]
1,4Sight^®^	0.2 mL/kg	7–8 °C	Russet Burbank	3	After a brief curing period	Swing fogger	330 days	[34]
1,4-DMN	0.04 mL/kg 0.01 mL/kg	4, 7, and 9 °C	Umatilla Russet Ranger Russet Russet Burbank	3	After curing	Thermal fog	200 days	[23]
1,4-DMN	0.056 mL/kg	15 +/2 °C	Russet Burbank Shepody FL1879 Russet Norkotah	1	Non-dormant/ slightly ‘peeping’ stage	Dribbling from a pipette onto gauze that was placed on top of the return air pipe.	66 days	[26]

**Table 3 plants-10-02307-t003:** Shelf-life studies on different levels of efficacy of *S*-carvone and Talent^TM^ as sprout suppressants.

Treatment	Dosage (mL/kg)	Temp.	Type of Cultivar	Application			Shelf-Life	Ref.
				Number	Stage	Method		
*S*-carvone	Bintje 0.6 Agria 0.6	5–7 °C	Bintjie Agria	After 42 days	After wound-healing	Swing fog apparatus	Bintje +15 days compared to CIPC. Agria 0 days compared to CIPC.	[49]
*S*-carvone	0.080	9 °C	Russet Burbank	Once after 112 days	Before the appearance of sprouts.	Fine mist	+70 days compared to control.	[4]
*S*-carvone	0.6	9.85 °C	Monalisa	Every 7 days	After curing	Regular sprinkling	+21 days compared to control.	[51]
*S*-carvone	0.6	5 °C, 10 °C 15 °C	Agria	24 times for 7 days	After curing	Wick freshener	5 °C +60 days compared to control. +0 days compared to CIPC. 10 °C +75 days compared to the control. +0 days compared to CIPC. 15 °C +90 days compared to the control. +15 days compared to CIPC.	[27]

**Table 4 plants-10-02307-t004:** Efficacy studies on SmartBlock^®^ as an alternative sprout suppressant.

Treatment	Dosage (mL/kg)	Temp.	Type of Cultivar	Application			Storage Period	Ref.
				Number	Stages	Method		
SmartBlock^®^	0.115	6 °C 9 °C	Saturna Russet Burbank	1 application 4 applications (After every 42 days)	After curing	Cyclomatic fogging system	84 days 168 days	[37]
SmartBlock^®^	0.1 and 0.3	4 °C 7.5 °C	Binje Monalisa Nicola	1 application 3 applications (After every 56 days)	After 25% of the shorter dormancy tubers started sprouting	Thermal (hot) fogging	168 days	[54]
SmartBlock^®^	0.115	21 °C	Cultivar type not specified.	1 application	Over 90% of the potato tubers had sprouted being 5 cm to 10 cm and as long as 25 cm.	Thermal fogging	21 days	[56]

**Table 5 plants-10-02307-t005:** Shelf-life studies on the efficacy of caraway seed essential oil as an alternative sprout suppressant.

Treatment	Dosage (mL/kg)	Temp. (°C)	Type of Cultivar	Application			Shelf-Life	Ref.
				Number	Stage	Method		
Caraway seeds		9.85	Monalisa	Every 7 days	After curing	Homogenous distribution	0 days compared to the control.	[51]
Caraway seed oil	155	8	Agria Kennebec	1	After curing	Vapor inside a box with a filter paper.	25 days out of 70 days.	[20]
Caraway and Dill essential oils	Caraway—0.048 * (0.96 mL/20 kg) Dill—0.035 * (0.69 mL/20 kg) CIPC—0.02 * (0.4 mL/20 kg)	5 10 15	Agria	24 times for 7 days	After curing	Wick freshener	5 °C +90 ± 60 ** days compared to control. +30 ± 0 ** days compared to CIPC. 10 °C +135 ± 90 ** days compared to control. +60 ± 15 ** days compared to ClPC. 15 °C +150 ± 105 ** days compared to control. +75 ± 30 ** days compared to CIPC.	[27]

* Recalculated from the reported data, which is in the bracket. ** Data represent Caraway/Dill.

## Data Availability

Data is contained within this article.

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
