# Peer review of "Evaluating Ecologically Acceptable Sprout Suppressants for Enhancing Dormancy and Potato Storability: A Review"

_plants, 2021, doi:10.3390/plants10112307_

Round 1

Reviewer 1 Report

Dear Authors,

The work entitled: “Evaluating ecologically acceptable sprout suppressants for en- 2 hancing dormancy and potato storability: a review” is well organized, The authors made a wide review of the literature. The work has many advantages, but it should be supplemented with several important elements:
1) In the introduction, the purpose of the research should be clearly specified.
2) Bearing in mind the aim of the research, the Authors should then specify the research hypothesis / or hypotheses and verify them later in the work.
3) There is no specific research methodology. Authors should provide information about the principles of collecting literature, and what type of literature sources were analyzed in the work. The point here is whether the authors conducted a systematic literature review of scientific articles indexed in databases such as Scopus, Web of Science, ScienceDirect, using the preferred reporting items for systematic reviews, treating them as guidelines. Whether or not other databases, such as Google Scholar and ResearchGate, were used or disregarded.
4) In this review, the authors have adopted the broader term for the associated with inhibition of germination and you should stick to it.
5) The most important terms used in the work should be very briefly defined and the abbreviations given in the list of abbreviations.
6) The lack of explanation of the phosiological and biochemical mechanisms of potato germination - this aspect should be completed.
7) Conclusions should be generalizing and summarizing and at the same time should be a response to the aim of the work adopted in the introduction.
8) Supplement the literature with important items related to natural growth inhibitors.

Author Response

Reviewer 1
1. In the introduction, the purpose of the research should be clearly specified.
Response: We have revised the last paragraph to make the purpose of the review clearer by succinctly stating the questions we sort to address with this review.
2. Bearing in mind the aim of the research, the Authors should then specify the research hypothesis / or hypotheses and verify them later in the work.
Response: As noted above, rather than providing hypothesis, we enumerated the core research questions and concepts that were discussed in the review in the introduction section while also editing the conclusions to ensure a clearer highlight of our findings from the literature review.
3. There is no specific research methodology. Authors should provide information about the principles of collecting literature, and what type of literature sources were analyzed in the work. The point here is whether the authors conducted a systematic literature review of scientific articles indexed in databases such as Scopus, Web of Science, ScienceDirect, using the preferred reporting items for systematic reviews, treating them as guidelines. Whether or not other databases, such as Google Scholar and ResearchGate, were used or disregarded.
Response: A concise research methodology has been included as suggested.
4. In this review, the authors have adopted the broader term for the associated with inhibition of germination and you should stick to it.
Response: We used sprouting to denote the germination of potato tubers and we have checked to ensure that this was used consistently.
5. The most important terms used in the work should be very briefly defined and the abbreviations given in the list of abbreviations.
Response: A list of abbreviation has been provided just after the keywords.
6. The lack of explanation of the physiological and biochemical mechanisms of potato germination - this aspect should be completed.
Response: We believe this is a very broad subject that delves into several other endogenous and exogenous cues that interact to regulate tuber sprouting, hence, it is generally outside the scope of this review. This is why we provided a concise review of the mechanism of action of each sprouting suppressant to give insight on where they feature in affecting sprout growth and development.
7. Conclusions should be generalizing and summarizing and at the same time should be a response to the aim of the work adopted in the introduction.
Response: The conclusions have been revised, to the best of our ability, to ensure that our generalisations and summaries of findings from extant literature are clearly and succinctly presented.
8. Supplement the literature with important items related to natural growth inhibitors.
Response: We are not very certain of what the review was trying to ask us to do here, however, we checked to ensure that relevant aspects of storage conditions, application methods, dosage, and cultivar types with significant bearing on tuber suppressant efficacies and the search for natural and eco-friendly alternatives were addressed to the best of our knowledge and extant literature.

Reviewer 2 Report

The paper presents a comparative review of different products used as sprouts inhibitors in long term potato storage. The topic is of interest to warrant publication. The review points out a ‘genotype x chemical’ interaction effect observed in some studies. The manuscript needs to be reviewed to improve the quality of the presentation. All Tables could be combined into one without losing the quality of information. The title and certain passages in the text appear to suggest that sprouts suppressants prolong the physiological dormancy of tubers whereas their effect is to inhibit sprout growth after dormancy breakage. Certain references need to be reviewed for their relevance as supporting evidence for the conveyed information. For example, references cited in L 30-33  section referring to potato nutritional quality have nothing to do that topic

Author Response

Reviewer 2

The paper presents a comparative review of different products used as sprouts inhibitors in long term potato storage. The topic is of interest to warrant publication. The review points out a ‘genotype x chemical’ interaction effect observed in some studies. The manuscript needs to be reviewed to improve the quality of the presentation. All Tables could be combined into one without losing the quality of information. The title and certain passages in the text appear to suggest that sprouts suppressants prolong the physiological dormancy of tubers whereas their effect is to inhibit sprout growth after dormancy breakage. Certain references need to be reviewed for their relevance as supporting evidence for the conveyed information. For example, references cited in L 30-33 section referring to potato nutritional quality have nothing to do that topic.

Response: We express our appreciation and gratitude for your observations and helpful criticism of our work. In consideration of the reviewer’s suggestion, we did a general check and revision of the whole manuscript to improve the presentation quality to the best of our ability. Sentences that suggested sprout suppressants extend physiological dormancy of tubers rather than inhibit sprout growth after dormancy breakage were also checked and have been revised where necessary to convey the information correctly. We believe that confusion must have arose, partly, from our use of the phrase sprouting suppression instead of sprout suppression. We understand the concerns of the reviewer with regards to having a single table as against several ones, however, the column heading of the tables are not completely uniform hence it would be challenging to have one table with all the information.

Regarding incorrect citations and references, we checked through the entire manuscript to ensure the correct references have been used and adjustments were made where necessary including the specific one mentioned by the reviewer. Again, we express our appreciation for your insightful and helpful criticism of our work.

Round 2

Reviewer 1 Report

For Authors

Dear Authors
The authors corrected and enriched the work with a detailed explanation of the purpose of the work, supplementing the research methodology, supplementing the application, conclusions and improving the English language, as recommended by the Reviewer. Thanks to this, the work will gain in quality.

Author Response

We are grateful to the reviewer for the careful consideration of the manuscript and for the constructive comments provided.